civil engineering/engineering geology/energy

roadway driving along adjacent goaf, narrow coal pillar, dynamic destabilization, rock burst mechanism, fold catastrophe model

**Author for correspondence:**
Zhengzheng Cao
e-mail: caozz@hpu.edu.cn

# Destabilization and energy characteristics of coal pillar in roadway driving along gob based on rockburst risk assessment

## Yi Xue[1], Zhengzheng Cao[2] and Wenlong Shen[2]

[1]Institute of Geotechnical Engineering, State Key Laboratory of Eco-hydraulics in Northwest Arid Region, Xi'an University of Technology, Xi'an 710048, People's Republic of China
[2]International joint research laboratory of Henan province for underground space development and disaster prevention, Henan Polytechnic University, Jiaozuo 454003, People's Republic of China

YX, 0000-0001-7728-1531; ZC, 0000-0003-0752-9903

Roadway driving along adjacent goafs is an effective method to develop the recovery rate of coal resources. However, rock burst triggered by dynamic destabilization of coal pillars poses a serious threat to safe and efficient mining, thereby significantly restricting the sustainable development of coal mines. In this study, from the perspectives of energy accumulation and dissipation, a mathematical model of coal pillars is established and the energy equilibrium relationship of the mechanical system is analysed. The rock burst mechanism of coal pillars in gob-side entry is obtained based on a fold catastrophe mathematical model. Results indicate that the rock burst triggered by the instability is a destabilization phenomenon. If the stiffness factor of the mechanical system is less than 1 and the external force is enough to lead coal pillars to the peak stress point, then rock burst disaster occurs. The engineering analysis and numerical simulation are conducted to study the rock burst in the gob-side entry that occurred in Xin'an coal mine. Stress release caused by mining can reduce the risk of rock burst to a certain extent. The amount of elastic energy released is $6.4512 \times 10^7$ J, which is close to the observation data and verifies the correctness and rationality of the research method. The research result provides a theoretical basis and technical guidance for rock burst prevention and control in roadway driving along adjacent goafs.

## 1. Introduction

The gob-side entry driving technique, which excavates the roadway along adjacent stable gobs with narrow coal pillars (approx. 3–8 m),

has been extensively employed and developed in Chinese mines in recent years to speed up the recovery rate of coal resources and roadway excavation [1–3]. This technique is a high-recovery and advanced mining technology that has been widely applied [4–6]. The stability of overburden strata is significantly influenced by the stabilization of coal pillars, which is vital to the improvement of coal recovery rate [7–12]. Thus, such stability is a key factor for sustainable production in coal mines.

Advances in the gob-side entry driving technique over the last 30 years include numerous research approaches and conclusions about coal pillar stabilization. On the basis of the movement characteristics of gob-side entry, Hou & Li [13] proposed the stabilization principles of primary and secondary structures, which indicate that improving the pre-tension of bolting and support strength is a key factor to guarantee stability. Bai *et al.* [14] suggested that coal pillars reinforced by high-strength bolts are important to the stability of surrounding rocks. Li *et al.* [15] established a roadway mechanical model that drives along adjacent gobs. On the basis of numerical simulation, theoretical analysis and engineering practice, Zheng *et al.* [16] proposed that the influences of coal mining and roadway driving should be considered simultaneously when the rational size of coal pillars is determined. Wang *et al.* [17] indicated three failure modes of basic roof in gob-side entry and researched the influence rule of failure structure on coal pillar stability. Li *et al.* [18] investigated six coal mines in the western and eastern parts of China to analyse and summarize the key factors that affect the deformation and failure model of coal pillars and suggested controlling countermeasures for different influencing factors to guarantee coal pillar stability. The above research results indicate that coal pillars are important elements in gob-side entry and make significant contributions to the promotion of the mining technique.

According to previous viewpoints, a roadway driving along adjacent goafs is excavated after the stable motion of overburden strata above the adjacent panels [19–22]. Therefore, the low-stress environment of the roadway that drives along goafs is beneficial for eliminating the risk of rock burst [23–25]. The gob-side entry driving technology has been widely adopted; consequently, the rock burst induced by coal pillars has initiated in the gob-side entry in recent mining practices, thereby posing severe threats to safe production [26–28]. Dou *et al.* [29] established the spatial structure evolution model of overlying strata to reveal this type of rock burst mechanism and suggested that this type of rock burst, which is called spatial structure instability rock burst, is related to the space structure of overburden strata in adjacent gobs, thereby establishing the research foundation for the rock burst mechanism of coal pillars. However, the dynamic instability mechanism of coal pillars is still not fully understood; it is the key problem in coal pillar rock burst and should be studied thoroughly and systematically.

Rock burst is the swift release process of elastic energy stored in coal rocks [30,31]. On the basis of catastrophe theory, Pan & Wang [32] conducted research on the rock burst mechanism of circular tunnels and indicated that rock burst occurs suddenly if the elastic deformation energy unleashed by the relatively undamaged part surpasses the dissipated energy.

Therefore, studying the rock burst triggered by the dynamic destabilization of coal pillars from the viewpoints of energy accumulation and dissipation is reasonable and feasible. In this study, we establish a mechanical system model of coal pillar and roadway roof in gob-side technology and analyse the energy equilibrium relationship in the mechanical system. On this basis, the equilibrium equation of the mechanical system is derived, and the rock burst mechanism of coal pillars is obtained. The research results indicate that the rock burst induced by coal pillars could be explained by the fold catastrophe mathematical model, which provides theoretical guidance for rock burst prevention and control in a roadway that drives along adjacent goafs.

# 2. Destabilization mechanism of coal pillars in gob-side entry

## 2.1. Basic principle of gob-side entry

Reasonable width design of coal pillars is the key to ensuring stability of surrounding rock in the roadway driving along the adjacent stable gob. According to the geological and mining conditions, the width optimization of coal pillars in gob-side entry is obtained on the basis of the abutment pressure distribution characteristics and the ultimate balanced theory in theoretical analysis. Unlike a sectional pillar (approx. 20–35 m) designed between adjacent working faces to separate mining disturbance influences (including the lateral stress reduction and stress superposition areas) in traditional longwall mining, a narrow pillar (approx. 4–10 m) is typically located in the reduction zone of lateral stress when the roadway is excavated along the boundary of adjacent gobs, and the overburden movement and stress redistribution tend toward a stable state in gob-side entry. Thus, a certain timespan (probably half

(a)

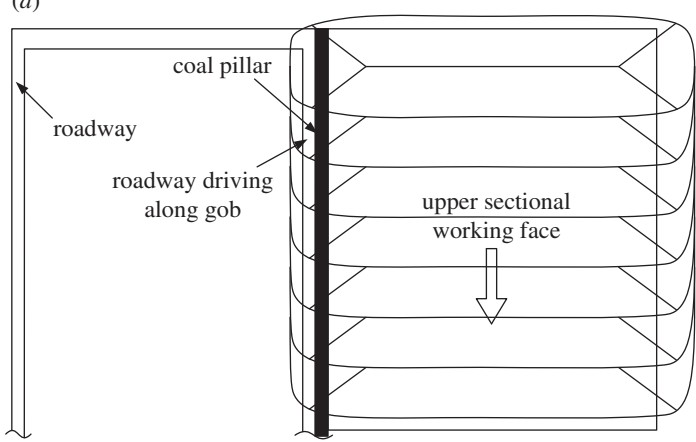

coal pillar

roadway

roadway driving
along gob

upper sectional
working face

(b)

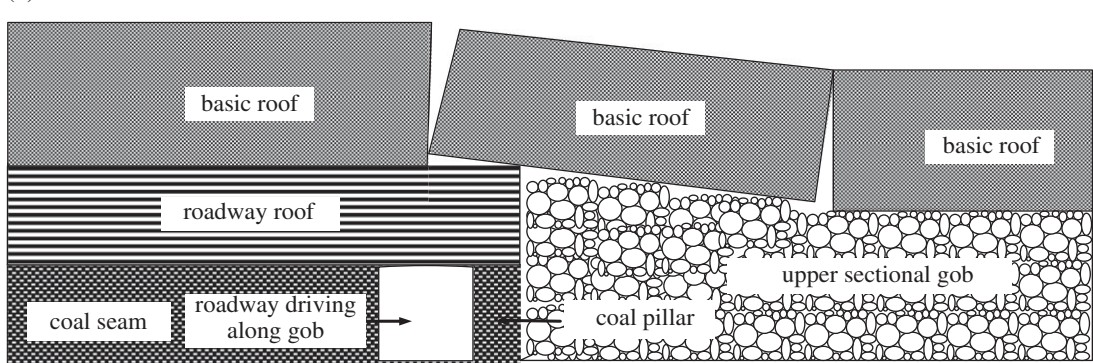

basic roof

basic roof

basic roof

roadway roof

upper sectional gob

coal seam

roadway driving
along gob

coal pillar

**Figure 1.** Mechanical model of a roadway driving along goafs. (*a*) Plane view and (*b*) sectional view.

a year) usually exists for the movement of overlying strata and stress redistribution after the adjacent longwall working face is finished. The structural mechanical model is shown in figure 1.

## 2.2. Mechanical model of roadway roof and coal pillar

The mechanical model of roadway roof and coal pillar is established according to the basic principle of a roadway driving along adjacent goafs. In the mechanical system model, the roadway roof is viewed as an elastic mass on the basis of its elastic behaviour, Hence, the load–displacement curve of the roadway roof is obtained as a straight line on, in which its slope is the stiffness factor $k_n = \tan \alpha$, as shown in figure 2.

In the uniaxial compression test of coal sample, when the working load reaches the compressive strength of the coal sample, the capability of the coal sample to resist deformation weakens with the increment in compression amount. The property of the coal sample is known as strain softening. The Weibull distribution law is introduced to describe a representative elemental volume of coal mass, and the load–displacement relationship of coal mass in the uniaxial compression test is presented by a smooth curve with a point of inflection $e$. Therefore, in combination with the damage theory of micro-statistics, the load–displacement relationship of coal mass is obtained as

$$F(u) = \lambda u \exp\left[-\left(\frac{u}{u_c}\right)^m\right], \tag{2.1}$$

where $\lambda$ represents the initial stiffness factor, $u_c$ represents the maximum displacement, and $m$ represents the kin exponent.

If the equation $F''(u_e) = 0$ is satisfied, then the displacement $u_e$ at the inflection point $e$ is obtained as

$$F''(u_e) = \frac{\lambda m}{u_c}\left[m\left(\frac{u_e}{u_c}\right)^{2m-1} - (1+m)\left(\frac{u_e}{u_c}\right)^{m-1}\right]\exp\left[-\left(\frac{u_e}{u_c}\right)^m\right] = 0. \tag{2.2}$$

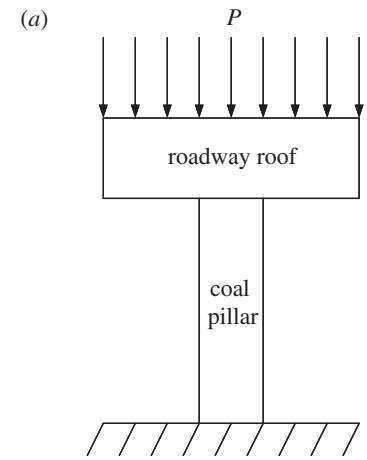

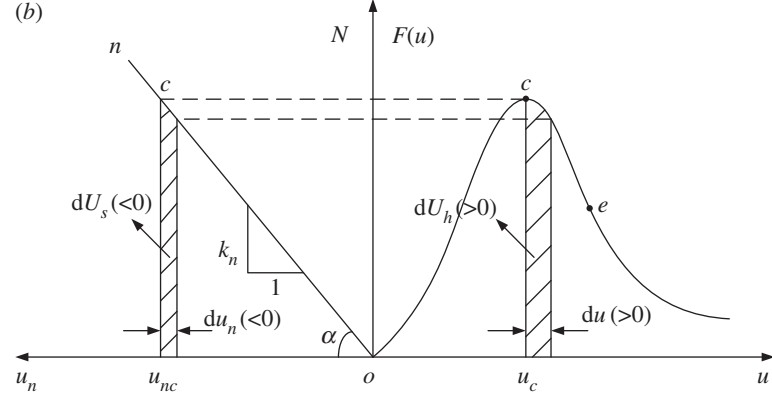

**Figure 2.** Mechanical model of roadway roof and coal pillar. (*a*) Interaction relationship of pillar-roof system and (*b*) load-displacement curve of mechanical system.

Thus, the relationship between $u_e$ and $u_c$ is acquired as

$$\frac{u_e}{u_c} = \left(\frac{1+m}{m}\right)^{1/m}. \tag{2.3}$$

When the displacement of the coal pillar is $u_c$, the right triangle area $ou_{nc}c$ is the energy reserved in the roadway roof and the curved line triangle $ou_cc$ is the energy reserved in the coal pillar, depicted in figure 2*b*. The difference between soft coal and hard coal is the curve slope of the load–displacement relationship of coal mass. When the incremental value of coal pillar displacement is $du(>0)$ in the strain-softening stage, the dissipated energy of coal pillars for plastic strain is $dU_h = F(u)du(>0)$; correspondingly, the incremental quantity of roadway roof displacement is $du_n(<0)$, and the unleashed energy of roadway roof is $dU_s = Ndu_n(<0)$. If the incremental quantity of roadway roof displacement, then $-Ndu_n < F(u)du$ or $-dU_s < dU_h$ is true, which illustrates that the complementary energy $dW = Pdu_P(>0)$ by external load $P$ is needed for the quasi-static deformation of coal pillars. The quasi-static deformation is the mechanical response of coal pillar under the action of quasi-static loading, namely, the term of inertia force caused by deformation is omitted.

## 2.3. Energy mechanism of the dynamic destabilization of coal pillars

The energy variation regularity in the mechanical model of roadway roof and coal pillars is shown in figure 3, where points $j$ and $s$ denote the initial point and terminal point in the coal pillar instability stage, respectively. The slope of the post-peak stage of the coal sample in $j$ and $s$ points are equal to that of the oblique one. Thus, the incremental quantity $dU_h$ of the dissipated energy of the coal pillar is less than the incremental quantity $dU_s$ of the unleashed energy of the roadway roof and the excess energy translated into the kinetic energy of the coal pillar, thereby leading to dynamic instability of the coal pillar.

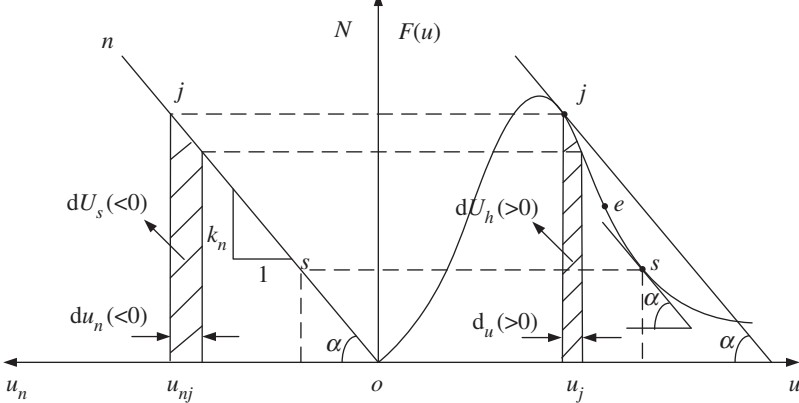

**Figure 3.** Energy variation regularity in the mechanical model of roadway roof and coal pillar.

The amount of unleashed energy used in the mechanical model is shown in figure 4, where the unleashed energy of the roadway roof is represented by the trapezoidal area $ju_{nj}u_{ns}sj$ from the initial point $j$ to the terminal point $s$, and the dissipated energy of the coal pillar is represented by the area $ju_ju_ssj$ of the trapezoid with a curved side. On this basis, the shaded area in figure 4, which subtracts the area $ju_{nj}u_{ns}sj$ from the area $ju_ju_ssj$, represents the excess energy of the mechanical model. The excess energy is transmitted to kinetic energy $\Delta E$ of the coal pillar.

# 3. Instability mechanism of coal pillar

## 3.1. Basic control equation of energy equilibrium

Through the analysis of elastic energy and the potential energy applied by external load, the energy summation in the mechanical model is obtained by the following equation:

$$\Pi = \frac{1}{2}k_nu_n^2 + \int_0^u F(u)du - \int_0^{u_P} P(u_P)\mathrm{d}u_P, \tag{3.1}$$

where $u_n$ and $u_P$ are related to $u$, and the equation $u_P = u_n + u$ is satisfied. On this basis, the derivation of total potential energy is obtained as

$$\frac{\mathrm{d}\Pi}{\mathrm{d}u} = k_nu_n\frac{\mathrm{d}u_n}{\mathrm{d}u} + F(u) - P(u_P)\frac{\mathrm{d}u_P}{\mathrm{d}u}. \tag{3.2}$$

In addition, on the basis of Newton's third law, the relationship between external load $P$ and internal force $N$ in roadway roof and internal force $F(u)$ in coal pillar is obtained as

$$P = N = k_nu_n = F(u). \tag{3.3}$$

Combining equation (3.2) with equation (3.3) leads to

$$\frac{\mathrm{d}\Pi}{\mathrm{d}u} = k_nu_n\frac{\mathrm{d}N}{k_n\mathrm{d}u} + F(u) - P\frac{\mathrm{d}u_P}{\mathrm{d}u} = F(u)\frac{F'(u)}{k_n} + F(u) - P\frac{\mathrm{d}u_P}{\mathrm{d}u} = 0. \tag{3.4}$$

Equation (3.4) is the energy–work balance equation in the mechanical model of roadway roof and coal pillar. The mathematical treatment in equation (3.4) is clarified as follows:

For the first term,

$$F(u)\frac{F'(u)}{k_n} = k_nu_n\frac{\mathrm{d}N}{k_n\mathrm{d}u} = N\frac{\mathrm{d}u_n}{\mathrm{d}u} = \frac{\mathrm{d}U_s}{\mathrm{d}u}. \tag{3.5}$$

For the second term,

$$F(u) = \frac{F(u)\mathrm{d}u}{\mathrm{d}u} = \frac{\mathrm{d}U_h}{\mathrm{d}u}. \tag{3.6}$$

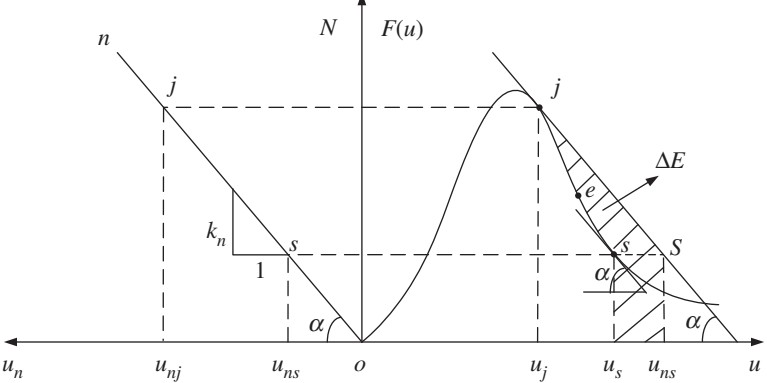

**Figure 4.** Unleashed energy amount of the mechanical model.

For the third term,

$$P(u_P)\frac{\mathrm{d}u_P}{\mathrm{d}u} = \frac{\mathrm{d}W}{\mathrm{d}u}. \tag{3.7}$$

The third term $\mathrm{d}W/\mathrm{d}u$ is denoted as $J$, referred to as 'the ratio of energy input', which is the energy applied by external load. Equation (3.4) is then rewritten as

$$F(u)\frac{F'(u)}{k_n} + F(u) - J = 0. \tag{3.8}$$

## 3.2. Mathematical model of the dynamic instability of coal pillar

The terms of $F(u)$ and $F(u)F'(u)$ in equation (3.4) are performed at the inflection point according to the Taylor expansion in mathematical analysis theory. Then,

$$\left(\frac{u-u_e}{u_e}\right)^2 + 2\left(\frac{u-u_e}{u_e}\right)\frac{(1-K)[F'(u_e)]^2}{u_e F(u_e)F'''(u_e)} + \frac{2(1-K)F'(u_e)}{u_e^2 F'''(u_e)} - \frac{2k_n J}{u_e^2 F(u_e)F'''(u_e)} + o((u-u_e)^2) = 0, \tag{3.9}$$

where $K = -k_n/F'(u_e)$ represents the stiffness factor of the mechanical model. On the basis of the various catastrophe models in mathematical theory, equation (3.9) belongs to the fold catastrophe model. When overlooking the higher-order value

$$\left\{\frac{u-u_e}{u_e} + \frac{(1-K)[F'(u_e)]^2}{u_e F(u_e)F'''(u_e)}\right\}^2 - \left\{\frac{(1-K)[F'(u_e)]^2}{u_e F(u_e)F'''(u_e)}\right\}^2 + \frac{2(1-K)F'(u_e)}{u_e^2 F'''(u_e)} - \frac{2k_n J}{u_e^2 F(u_e)F'''(u_e)} = 0. \tag{3.10}$$

Equation (2.1) is substituted into equation (3.10), and the balance control relationship in the mechanical model of roadway roof and coal pillar is obtained as

$$\left[\frac{u-u_e}{u_e} + \frac{m(1-K)}{(1+m)^2}\right]^2 - \left[\frac{m(1-K)}{(1+m)^2}\right]^2 - \frac{2(1-K)}{(1+m)^2} - \frac{2KJ}{F(u_e)(1+m)^2} = 0, \tag{3.11}$$

where $K = -k_n/F'(u_e) = k_n/\lambda m \exp(-(1+m)/m)$.

The variable is substituted as follows:

$$\left.\begin{aligned} x &= \frac{u-u_e}{u_e} + \frac{m(1-K)}{(1+m)^2} \\ a &= -\left[\frac{m(1-K)}{(1+m)^2}\right]^2 - \frac{2(1-K)}{(1+m)^2} - \frac{2KJ}{F(u_e)(1+m)^2} \end{aligned}\right\} \tag{3.12}$$

Equation (3.11) can be expressed as

$$x^2 + a = 0. \tag{3.13}$$

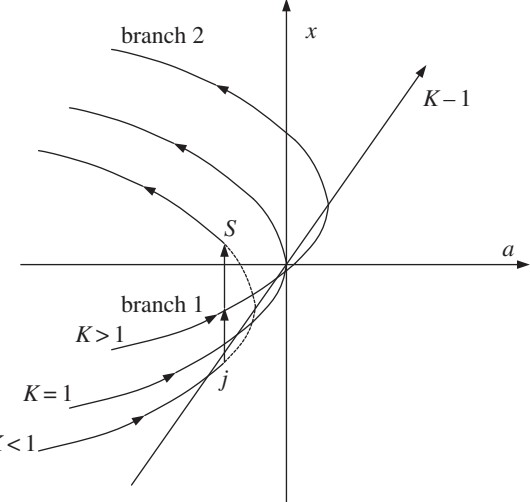

**Figure 5.** Balance curve of the fold catastrophe model.

Thus, the state variable $x$ is obtained as

$$\left.\begin{aligned}
x_1 &= \frac{u - u_e}{u_e} + \frac{m(1 - K)}{(1 + m)^2} = -\sqrt{-a} = -\sqrt{\left[\frac{m(1 - K)}{(1 + m)^2}\right]^2 + \frac{2(1 - K)}{(1 + m)^2} + \frac{2KJ}{F(u_e)(1 + m)^2}} \\
\text{and}\quad x_2 &= \frac{u - u_e}{u_e} + \frac{m(1 - K)}{(1 + m)^2} = \sqrt{-a} = \sqrt{\left[\frac{m(1 - K)}{(1 + m)^2}\right]^2 + \frac{2(1 - K)}{(1 + m)^2} + \frac{2KJ}{F(u_e)(1 + m)^2}}.
\end{aligned}\right\} \tag{3.14}$$

The fold catastrophe model is one of the seven types of catastrophe mathematical model, as shown in figure 5. The point in equilibrium surface refers to an equilibrium state of the mechanical system and the equilibrium surface of the fold catastrophe model. Two equilibrium states exist for the fold catastrophe model, namely, unstable and stable. One equilibrium state cannot change to the other equilibrium state.

## 3.3. Released energy of the mechanical system

Points $j$ and $s$ are the critical instability states in the balance curve of the fold catastrophe model. Hence, the value of $J$ is zero, i.e. $J(u_j) = J(u_s) = 0$, the control variable $a_j = a_s$. $J(u_j) = J(u_s) = 0$ is substituted into equation (3.14), and the expressions of state variable $x$ in points $j$ and $s$ are obtained as

$$\left.\begin{aligned}
x_j &= \frac{u_j - u_e}{u_e} + \frac{m(1 - K)}{(1 + m)^2} = -\sqrt{-a_j} = -\sqrt{\left[\frac{m(1 - K)}{(1 + m)^2}\right]^2 + \frac{2(1 - K)}{(1 + m)^2}} \\
\text{and}\quad x_s &= \frac{u_s - u_e}{u_e} + \frac{m(1 - K)}{(1 + m)^2} = \sqrt{-a_s} = \sqrt{\left[\frac{m(1 - K)}{(1 + m)^2}\right]^2 + \frac{2(1 - K)}{(1 + m)^2}}.
\end{aligned}\right\} \tag{3.15}$$

With the integration of equation (3.13), the non-dimensional potential energy is

$$\Pi_o = \frac{1}{3}x^3 + ax \tag{3.16}$$

and

$$\left.\begin{aligned}
\Pi_{oj} &= \frac{1}{3}x_j^3 + a_j x_j \\
\Pi_{os} &= \frac{1}{3}x_s^3 + a_s x_s,
\end{aligned}\right\} \tag{3.17}$$

where $a_j = a_s$, and $a = a_j$, $x_1 = x_j$, $x_2 = x_s$ are set in equations (3.13) and (3.14). $J = 0$ is set in $a_j$, $a_s$, $x_j$ and $x_s$. Accordingly, the non-dimensional energy released by the mechanical model of roadway

roof and coal pillar is

$$E_o = \Delta\Pi_o = \Pi_{oj} - \Pi_{os} = \frac{1}{3}(x_j^3 - x_s^3) + a_j(x_j - x_s).$$ (3.18)

In figure 5, $x_j = -x_s$, $a_j = -x_j^2 = -x_s^2$, $x_s > 0$. Equation (3.18) is rewritten as

$$E_o = \Delta\Pi_o = \frac{4}{3}x_s^3 = \frac{4}{3}\sqrt[\frac{3}{2}]{\frac{m^2(1-K)^2}{(1+m)^4} + \frac{2(1-K)}{(1+m)^2}}.$$ (3.19)

# 4. Engineering analysis and numerical simulation

## 4.1. Geological and mining conditions

A rock burst induced by coal pillar in the gob-side entry, located in the frontal abutment pressure region of No. 3306 working face, occurred in Xin'an coal mine, Shandong Province, China. The earthquake observation station in the Mining Group records that the rock burst intensity is equivalent to an earthquake of 2.3 Richter magnitude.

The average burial depth of coal seam is 550 m, with the average thickness of 5.05 m. The coal mass is a banded structure with fracture development, a hardness factor of $f = 3.3$ (Mohs scale) and a uniaxial compressive strength of 14 MPa. The immediate roof is sandy mudstone, with a thickness of 8.5 m, a hardness factor of $f = 5.9$ (Mohs scale) and a uniaxial compressive strength of 36 MPa. The immediate floor of the coal seam is siltstone with a thickness of 7.5 m, a hardness factor of $f = 5.2$ (Mohs scale) and a uniaxial compressive strength of 30 MPa. A hardness factor of $f = 7.3$ (Mohs scale) and a uniaxial compressive strength of 48 MPa. Those parameters are prepared for the theoretical calculation and numerical simulation.

The burst tendency is an inherent property of coal, and there are four indices of burst tendency (dynamic damage time, uniaxial compressive strength, elastic energy index and impact energy index), which can be obtained in the complete stress–strain curve in mechanical testing of bursting tendency of coal samples. The measurement results of the burst tendency indicators of coal samples indicate that the dynamic damage time is 46 ms, the elastic energy index is 6.79, the impact energy index is 5.14 and the uniaxial compressive strength is 14.0 MPa. The comprehensive evaluation results of burst tendency indicate that the coal samples show a strong impact tendency.

The roadway, which is driven along the stable gob of No. 3304 working face, is excavated as the ventilation roadway of No. 3306 working face. The dynamic destabilization of coal pillar leads to rock burst, as shown in figure 6. The rib spalling of the coal pillar is serious due to the large height of the roadway, and the coal wall moves 400–1000 mm, accompanied by loud noises and vibration. The bottom heave of the roadway floor occurs with a maximum value of 800 mm, and the steel ladder, cross beam and rock bolt are badly broken.

## 4.2. Establishment of numerical model

A mechanical experiment is conducted on the coal samples of the Xin'an mine, and the relevant mechanical data of coal samples are obtained as follows: peak stress $\sigma_c = 14$ MPa, elastic modulus $E = 4.47$ GPa, peak strain $\varepsilon_c = 5 \times 10^{-3}$, peak displacement $u_c = 2.5 \times 10^{-2}$ m, initial stiffness $\lambda = 4.48 \times 10^5$ MPa, systematical stiffness $K = 0.6$ and homogeneous index $m = 1$.

Taking the roadway of Xin'an mine as the engineering background, the distribution characteristics of supporting stress of the coal pillars after the roadway excavation are obtained using the numerical calculation software FLAC3D. The surrounding stress will change after the 7# coal seam is mined. According to the geological parameters of Xin'an mine, the relevant mechanical parameters and stratum thickness are determined. The periphery and bottom of the model is constrained by normal displacement. The failure of coal and rock caused by mining is evaluated by the Mohr–Coulomb failure criterion.

## 4.3. Evolution law of abutment stress in the front of the mining face

There is obvious damage and stress release ahead of the working face in the coal mining process. The strength of the coal body will be weakened greatly, and the cracks in the coal seam are widely developed. The energy accumulated in the coal body also has a larger release, and it will also reduce the impact tendency of the coal seam.

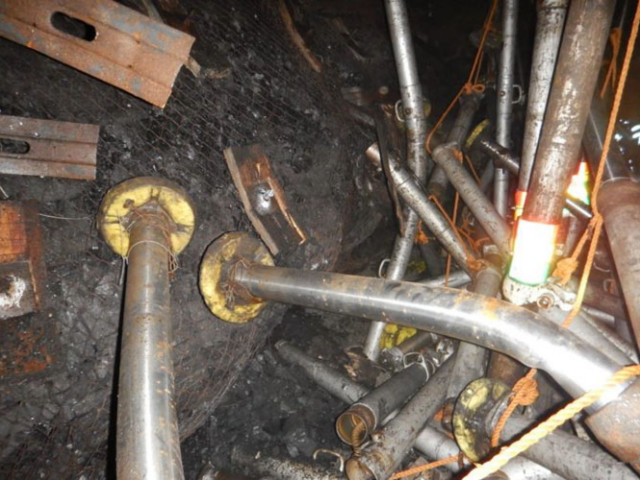

**Figure 6.** Site condition of the rock burst.

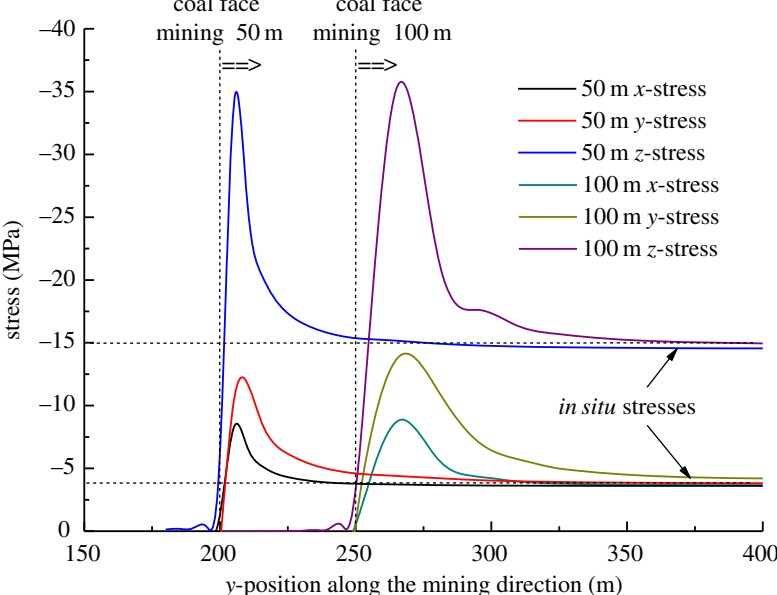

**Figure 7.** Stress distribution curves in front of the coal face when the coal seam 7# is mined 50 and 100 m.

Figure 7 is the stress distribution characteristic of the coal seam in front of the working face. It is obvious that the vertical stress of the coal seam is larger than that of horizontal stress, which is more than two times that of horizontal stress. When the coal seam is mined for 50 m, the stress concentration appears at 10 m, and the stress concentration points of three-dimensional stress appears in the same place. When the coal seam is mined for 100 m, the stress concentration appears at 40 m in front of the working face, and the stress concentration position is farther away from the working face. The maximum Z-direction stress is 35 MPa, and the energy accumulated in the coal seam at the stress concentration is larger, so rock burst accident occurs easily in the mining process.

## 4.4. Stress evolution of protected coal seam

Figure 8 shows distribution features of stress of the coal seam after finishing 7# coal seam. The stress in overlying strata is released after the mining of the coal seam. The stress concentration zone, the stress release zone and the original stress zone appear in the 3# coal seam. Besides, the minimum stress of the 3# coal seam is 4.6 MPa, and the minimum stress in the 2# coal seam is 8.2 MPa. The stress concentration area is not obvious in the 2# coal seam. It is also shown that there is a clear difference in the degree of stress release as the distance to the mining area varies. The nearer the distance, the

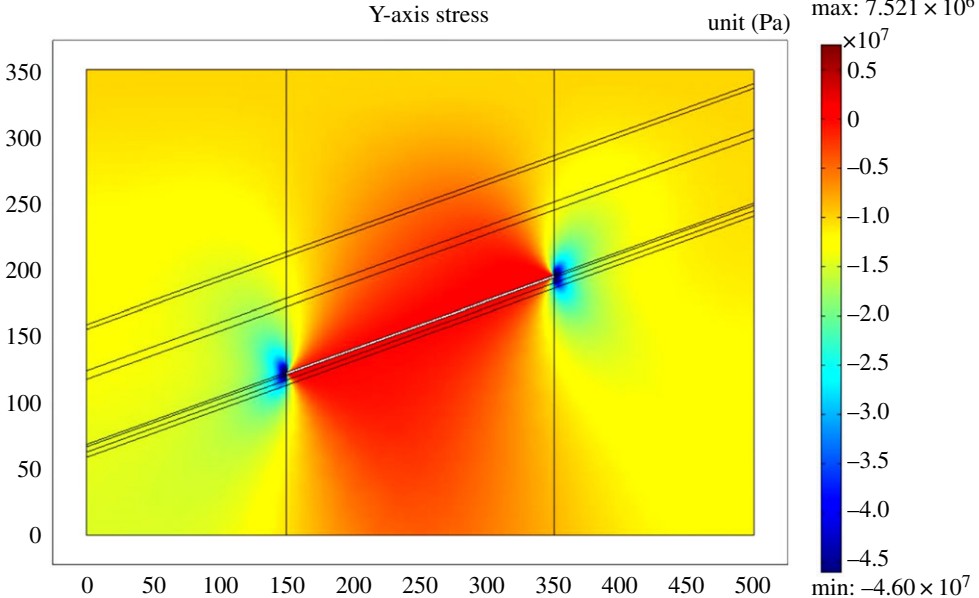

**Figure 8.** *Y*-axis stress in the coal seam.

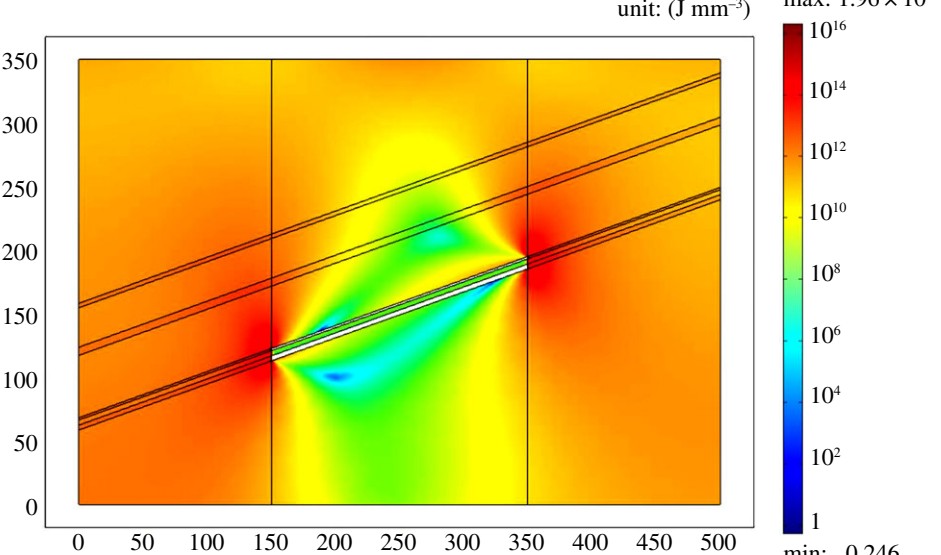

**Figure 9.** Energy density in the coal seam.

higher the stress release. At the same time, 3# and 2# coal seams also experienced a complicated stress evolution process in mining. The coal seam is in the original stress zone first. With the advance of the working face, the stress concentration appears and is then released.

## 4.5. Evolution of three-dimensional stress in the coal seam

After coal mining, there will be a big change in stress evolution of the coal seam. The horizontal stress distribution and vertical stress distribution of the coal seam are similar, but there are also some differences. The stress concentration of the Y direction is more obvious, and a distinct stress concentration area is formed at both ends of the goaf, indicating that the risk of potential rock burst is higher. A larger stress unloading zone appeared above and below the goaf, showing an elliptical shape. Figure 9 is a sketch map of the energy density distribution of the coal seam, which can more clearly analyse the energy storage characteristics of the coal mining process. There is an obvious low energy

density area at the upper and lower parts of the goaf, while the energy density is higher at the two ends of the goaf. Under the goaf, there is a blue arc area. The energy density of the region is the lowest area of the whole model, which shows that the stress of the arc area is fully released, and the danger of rockburst is greatly reduced. However, a large amount of energy is accumulated at both ends of the goaf, and there is still a large risk of rockburst. A greater understanding of stress and energy evolution of coal seam mining is beneficial for researching the energy characteristics of coal pillars in gob-side entry.

## 4.6. The evaluation of elastic energy released by mechanical system

Taking the roadway of Xin'an mine as the engineering background, the distribution characteristics of supporting stress of the coal pillars after the roadway excavation are obtained by the numerical calculation software.

$m = 1$ is substituted into equation (2.3), and the relationship of $u_e$ and $u_c$ is obtained as

$$u_e = \left(\frac{1+m}{m}\right)^{1/m} u_c = 2u_c = 5 \times 10^{-2}. \tag{4.1}$$

Equation (3.19) is combined with $m = 1$, and the quantity of dimensionless elastic energy released by the mechanical system is

$$E_o = \Delta\Pi_o = \frac{4}{3}\sqrt[\frac{3}{2}]{\frac{m^2(1-K)^2}{(1+m)^4} + \frac{2(1-K)}{(1+m)^2}} = \frac{4}{3}\sqrt[\frac{3}{2}]{\left(\frac{1-K}{4}\right)^2 + \frac{(1-K)}{2}}. \tag{4.2}$$

Equation (3.11) is combined with $m = 1$, and the dimensionless equilibrium equation of the mechanical system is

$$\left(\frac{u-u_e}{u_e} + \frac{1-K}{4}\right)^2 - \left(\frac{1-K}{4}\right)^2 - \frac{1-K}{2} - \frac{KJ}{2F(u_e)} = 0. \tag{4.3}$$

The dimensionless equilibrium equation (4.3) is compared with the dimensional equilibrium equation (3.8). On the basis of equation (4.2), the quantity of dimensional elastic energy released by the mechanical system is

$$T = \frac{2}{K}F(u_e)u_e\Delta\Pi_o. \tag{4.4}$$

The relevant mechanical data are substituted into equation (4.4) as

$$T = 6.4512 \times 10^7 \text{J}. \tag{4.5}$$

The intensity of this rock burst, in which the energy released is $T = 6.4512 \times 10^7$J, is equivalent to an earthquake of 2.1 Richter magnitude; therefore, the calculated value of the fold catastrophe model is close to the observation result. The released energy is sufficient to destroy coal pillars ranging from dozens of metres to over 100 m in the gob-side entry. Therefore, in the coal mining process, the excavation causes stress field redistribution in the original rock mass, and this phenomenon leads to stress concentration and elastic strain energy accumulation. It is obvious that the strength of a rockburst is proportional to the differential value of accumulated energy and dissipated energy in a coal mass.

## 5. Conclusion

(1) The rock burst triggered by the dynamic instability of the coal pillar is a physical destabilization phenomenon caused by the strain-softening characteristic exhibited by coal mass. If the incremental quantity of the dissipated energy of the coal pillar is less than the incremental quantity of the unleashed energy of the roadway roof in the strain-softening stage of coal mass and the excess energy is translated into the kinetic energy of the coal pillar, then the dynamic instability of the coal pillar occurs.

(2) The rock burst mechanism of coal pillars in the gob-side entry is obtained by establishing a mechanical model of roadway roof and coal pillar on the basis of the energy–work balance equation and the fold catastrophe mathematical model. When the stiffness factor of the

mechanical model is less than 1 and the external force is sufficient to lead the coal pillar to the peak stress point, the rockburst disaster occurs.

(3) The engineering analysis and numerical simulation are conducted to study the rockburst in the gob-side entry that occurred in Xin'an coal mine. Stress release caused by mining can reduce the risk of rockburst to a certain extent. The amount of elastic energy released by the mechanical system is $6.4512 \times 10^7$ J, equivalent to an earthquake of 2.1 Richter magnitude, which is close to the observation data and verifies the correctness and rationality of the research method.

(4) Reasonable development layout and protective seam mining technology are adopted in engineering practice to reduce the high stress concentration in the coal seam and prevent high load from leading the coal pillar to the peak stress point. Water injection in the coal seam is also employed to reduce the strain-softening stiffness factor of the coal pillar in the post-peak stage and improve the stiffness factor of a double-block system in rockburst prevention.

Data accessibility. Our data are from the filed case. The numerical parameters are based on geological data of Xin'an mine. The original geological data are deposited at the Dryad Digital Repository: http://dx.doi.org/10.5061/dryad.j5m024n [33].

Authors' contributions. Y.X. conceived the model; Z.C. performed the numerical simulation; Z.C. and Y.X. wrote the paper; W.S. and Y.X. revised the paper.

Competing interests. The authors declare no competing interests.

Funding. This work was supported by the National Natural Science Foundation of China (51804099), the Project Supported by Natural Science Basic Research Plan in Shaanxi Province of China (2019JQ-275), the China Postdoctoral Science Foundation (2019T120929 and 2018M633549), the Postdoctoral Research Projects of Shaanxi Province, the Scientific and Technological Development Project for Coal Mine Safety Production by Henan Province (HN19-59), the Foundation for Higher Education Key Research Project by Henan Province (19A130001), the Ph.D. Programs Foundation of Henan Polytechnic University (B2018-65).

Acknowledgement. The authors appreciate the anonymous reviewers for their constructive comments and suggestions.

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
