## [Reviewer comments · Royal Society Open Science]

Review History

RSOS-190094.R0 (Original submission)

Review form: Reviewer 1

Is the manuscript scientifically sound in its present form?

Yes

Are the interpretations and conclusions justified by the results?

Yes

Is the language acceptable?

Yes

Is it clear how to access all supporting data?

Not Applicable

Do you have any ethical concerns with this paper?

No

Have you any concerns about statistical analyses in this paper?

No

Recommendation?

Accept with minor revision (please list in comments)

Comments to the Author(s)

See attached file (Appendix A).

Review form: Reviewer 2

Is the manuscript scientifically sound in its present form?

No

Are the interpretations and conclusions justified by the results?

No

Is the language acceptable?

Yes

Is it clear how to access all supporting data?

Yes

Do you have any ethical concerns with this paper?

No

Have you any concerns about statistical analyses in this paper?

No

Recommendation?

Major revision is needed (please make suggestions in comments)

Comments to the Author(s)

The authors present a theoretical analysis of the energy accumulation and dissipation of coal pillar in gob-side entry. The numerical simulation and case study are conducted to verify the rationality of the research method. I have a few major comments as follows:

1. In section 2.2, the author used two mathematical models, named "Weibull distribution law" and "damage theory of micro-statistics". What is the basis for this? A theoretical model established needs to be based on a large number of experiments. The authors need to justify that these two models can truly describe the strain softening characteristics of coal samples. Otherwise, the results derived from such a non-consistent model are thus questionable.
2. What is "quasi-static deformation". Please explain it in detail.
3. How to determine the position of the points j and s in the coal pillar instability stage (Figure 3)?
4. The authors need to justify the reasonability of the fold catastrophe model compared to other previous studies.
5. In line 35 of P11, what is the criterion for judging the coal seam's burst tendency?
6. Page 13. Correct: Figures 11 and 12.
7. In the numerical modeling, I didn't see any results about crack propagation. How does the author come to this conclusion that "Many cracks in coal seam are development,"? (Please

see Wu et al., 2019. Rock Mechanics and Rock Engineering)

8. In figure 14, the energy density at the two ends of goaf reach to 1.96×10^{16} J. This value is so large. Is it reasonable? Please further analyze and discuss.

9. I didn't see the roadway driving along the goaf in the numerical model. Therefore, how does the author compare the theoretical results with the observation result (Line 14 to 16 of P15)? It is questionable.

10. The topic of paper is energy characteristics of coal pillar in gob-side entry. But, in the section of numerical simulation, the authors focuses on the analysis the stress evolution of coal seam mining rather than roadway driving along goaf. The numerical results need to be more closely related to the topic of the article.

Decision letter (RSOS-190094.R0)

17-May-2019

Dear Dr Xue,

The editors assigned to your paper ("Destabilisation and energy characteristics of coal pillar in roadway driving along gob based on rockburst risk assessment") have now received comments from reviewers. We would like you to revise your paper in accordance with the referee and Associate Editor suggestions which can be found below (not including confidential reports to the Editor). Please note this decision does not guarantee eventual acceptance.

Please submit a copy of your revised paper before 09-Jun-2019. Please note that the revision deadline will expire at 00.00am on this date. If we do not hear from you within this time then it will be assumed that the paper has been withdrawn. In exceptional circumstances, extensions may be possible if agreed with the Editorial Office in advance. We do not allow multiple rounds of revision so we urge you to make every effort to fully address all of the comments at this stage. If deemed necessary by the Editors, your manuscript will be sent back to one or more of the original reviewers for assessment. If the original reviewers are not available, we may invite new reviewers.

If your study uses humans or animals please include details of the ethical approval received, including the name of the committee that granted approval. For human studies please also detail

whether informed consent was obtained. For field studies on animals please include details of all permissions, licences and/or approvals granted to carry out the fieldwork.

- Data accessibility

If you wish to submit your supporting data or code to Dryad (<http://datadryad.org/>), or modify your current submission to dryad, please use the following link:
<http://datadryad.org/submit?journalID=RSOS&manu=RSOS-190094>

- Competing interests

- Authors' contributions

- Acknowledgements

- Funding statement

Kind regards,
Alice Power
Royal Society Open Science

on behalf of Dr Ian Moore (Associate Editor) and R. Kerry Rowe (Subject Editor)
openscience@royalsociety.org

Associate Editor's comments (Dr Ian Moore):

The manuscript has been assessed by two qualified reviewers. Each indicates that it may be publishable after revision. Therefore, I recommend you undertake revisions to address their concerns.

Comments to Author:

Reviewers' Comments to Author:

Reviewer: 1

Comments to the Author(s)

See attached file.

Reviewer: 2

Comments to the Author(s)

The authors present a theoretical analysis of the energy accumulation and dissipation of coal pillar in gob-side entry. The numerical simulation and case study are conducted to verify the rationality of the research method. I have a few major comments as follows:

1. In section 2.2, the author used two mathematical models, named "Weibull distribution law" and "damage theory of micro-statistics". What is the basis for this? A theoretical model established needs to be based on a large number of experiments. The authors need to justify that these two models can truly describe the strain softening characteristics of coal samples. Otherwise, the results derived from such a non-consistent model are thus questionable.
2. What is "quasi-static deformation". Please explain it in detail.
3. How to determine the position of the points j and s in the coal pillar instability stage (Figure 3)?
4. The authors need to justify the reasonability of the fold catastrophe model compared to other previous studies.
5. In line 35 of P11, what is the criterion for judging the coal seam's burst tendency?
6. Page 13. Correct: Figures 11 and 12.
7. In the numerical modeling, I didn't see any results about crack propagation. How does the author come to this conclusion that "Many cracks in coal seam are development,"? (Please see Wu et al., 2019. Rock Mechanics and Rock Engineering)
8. In figure 14, the energy density at the two ends of goaf reach to 1.96×10^{16} J. This value is so large. Is it reasonable? Please further analyze and discuss.
9. I didn't see the roadway driving along the goaf in the numerical model. Therefore, how does the author compare the theoretical results with the observation result (Line 14 to 16 of P15)? It is questionable.
10. The topic of paper is energy characteristics of coal pillar in gob-side entry. But, in the section of numerical simulation, the authors focuses on the analysis the stress evolution of coal seam mining rather than roadway driving along goaf. The numerical results need to be more closely related to the topic of the article.

Author's Response to Decision Letter for (RSOS-190094.R0)

See Appendix B.

RSOS-190094.R1 (Revision)

Review form: Reviewer 1

Is the manuscript scientifically sound in its present form?

Yes

Are the interpretations and conclusions justified by the results?

Yes

Is the language acceptable?

Yes

Do you have any ethical concerns with this paper?

No

Recommendation?

Accept as is

Comments to the Author(s)

I am happy with the revision. I suggest publish it as it is.

Review form: Reviewer 2

Is the manuscript scientifically sound in its present form?

Yes

Are the interpretations and conclusions justified by the results?

Yes

Is the language acceptable?

Yes

Do you have any ethical concerns with this paper?

No

Recommendation?

Accept as is

Comments to the Author(s)

The revised paper provides a better explanation of the energy characteristics of coal pillar in roadway driving along gob and answers the all questions. The revised manuscript has also

improved the clarity of presentation and language. This reviewer recommends its acceptance for publication.

Decision letter (RSOS-190094.R1)

18-Jun-2019

Dear Dr Xue,

I am pleased to inform you that your manuscript entitled "Destabilisation and energy characteristics of coal pillar in roadway driving along gob based on rockburst risk assessment" is now accepted for publication in Royal Society Open Science.

on behalf of Dr Ian Moore (Associate Editor) and R. Kerry Rowe (Subject Editor)
openscience@royalsociety.org

Reviewer comments to Author:
Reviewer: 1

Comments to the Author(s)
I am happy with the revision. I suggest publish it as it is.

Reviewer: 2

Comments to the Author(s)
The revised paper provides a better explanation of the energy characteristics of coal pillar in roadway driving along gob and answers the all questions. The revised manuscript has also improved the clarity of presentation and language. This reviewer recommends its acceptance for publication.

Appendix A

This manuscript presents an investigation on the mechanism of rockbursts in the context of gob-side entry roadway driving, which is of great interest to coal mining engineers. It is an important topic of rock mechanics and rock engineering and very interesting for the readers of the journal. This article is very innovative and worth publishing. However, a concern with the manuscript is the explanation of results. The numerical analysis of the results should be further described.

Therefore, I recommend publication of this manuscript after the minor revision. Specific comments and suggestions for the improvement of the manuscript are noted below.

1. Roadway driving along adjacent goafs is an effective method to develop the recovery rate of coal resources. The wide of coal pillar is important, the authors do not discuss. This part should be replenished in the revised version.
2. We usually describe the dispersion of rock materials by Weibull distribution, normal distribution or other distribution. In section 2.2, is it valid to use Weibull distribution to describe the F-u relationship of coal? The author should elaborate on this part.
3. In section 4.1, what are those parameters for?
4. In section 4.2, rock burst index is a very effective index to predict rock burst tendency. How are the dynamic damage time, elastic energy index and impact energy index defined and measured?
5. In section 4.4, the variation of stress abutment distance ahead of the working face (from 10 m at 50 m advance to 40 m at 100 m advance) should not be expected, in that the model which is homogeneous should yield similar volumes of failure zones around the working face.
6. The theory proposed is reasonable and innovative. Therefore, I recommend the use of theoretical models in numerical models and the authors can strengthen of the connection with the theoretical part.
7. Coal and gas outburst and rock burst are the most serious dynamic disasters in coal mining. In shallow mining, most of the dynamic disasters are single coal and gas outburst or rock burst, and the interaction and interaction are not significant. With the increase of mining depth, the interaction between the two kinds of disasters began to appear, resulting in coal and gas outburst, rock burst and two kinds of disasters coexisting and compounding each other. Both of them are unstable sudden dynamic disasters, and their triggering mechanism can be explained by energy theory. Whether the author tries to combine them?
8. In the process of mining, the effects of soft coal and hard coal seem to be very different, and their risk of rock burst is different. For the model of Fig. 2a, does the author consider the influence of strength on rock burst of coal body?
9. The mining stress path is a process in which the vertical stress increases and the horizontal stress decreases during the mining process, which is inconsistent with the conventional stress loading path. Therefore, can Figure 3 reflect the effect of mining process on rock stress and strain? I suggest that the author explain this matter more clearly in order to provide better guidance and help for mining.

Appendix B

Summary of amendments

Re: “**Destabilisation and energy characteristics of coal pillar in roadway driving along gob based on rockburst risk assessment**”

For **Royal Society Open Science**

Manuscript ID: RSOS-190094

Many thanks for reviewers and editor’s constructive comments. These comments and suggestions have been carefully incorporated into the revised manuscript. The point-to-point replies for the comments are summarized as below:

Reviewer #1:

This manuscript presents an investigation on the mechanism of rockbursts in the context of gob-side entry roadway driving, which is of great interest to coal mining engineers. It is an important topic of rock mechanics and rock engineering and very interesting for the readers of the journal. This article is very innovative and worth publishing. However, a concern with the manuscript is the explanation of results. The numerical analysis of the results should be further described.

Therefore, I recommend publication of this manuscript after the minor revision. Specific comments and suggestions for the improvement of the manuscript are noted below.

(1) Roadway driving along adjacent goafs is an effective method to develop the recovery rate of coal resources. The wide of coal pillar is important, the authors do not discuss. This part should be replenished in the revised version.

A: Thank you for this suggestion. Reasonable width design of coal pillar is the key for ensuring stability of surrounding rock in the roadway driving along adjacent stable gob. According to the geological and mining conditions, the width optimization of coal pillar in gob-side entry is obtained on the basis of the abutment pressure

distribution characteristics and the ultimate balanced theory in theoretical analysis. At present, the typical dimensions for narrow pillar width in various situations (such as in different mining depths, mining thickness, roadway widths, mining methods, strata spatial structures etc.) have not been summarized comprehensively in published literature. This issue is the next research topic for the authors. The related research results have been achieved by Bai et al. (2004) as follow: through the numerical calculation and analysis, the relationship among the stability of narrow coal pillar, the mechanical property of coal seam and the pillar width is studied. According to various conditions of the coal seams, the rational width of the corresponding narrow coal pillar is determined; the rational width is 4~5 m in soft coal seam, and the rational width is 3~4 m in medium-hard coal seam. This manuscript was revised with reference to this suggestion.

Reference:

[1] Bai JB, Hou CJ and Huang HF (2004) Numerical simulation study on stability of narrow coal pillar of roadway driving along goaf. *Chinese Journal of Rock Mechanics and Engineering* 23(20): 3475-3479.

2. We usually describe the dispersion of rock materials by Weibull distribution, normal distribution or other distribution. In section 2.2, is it valid to use Weibull distribution to describe the F-u relationship of coal? The author should elaborate on this part.

A: Thank you for this suggestion.

The dispersion of rock materials has an important influence on the results of research question, and many distribution types (such as Weibull distribution, normal distribution or other distribution) could be employed to clarify this characteristic of rock materials. Many experiments (such as Karcinovic D. (1982); Pan, Y. (2004) et al.) have been done to clarify that "Weibull distribution law" and "damage theory of micro-statistics" are suitable for the dispersion and strain softening characteristics of of rock materials. On this basis, it is valid to use "Weibull distribution law" and "damage theory of micro-statistics" to describe the F-u relationship of coal, and the

load–displacement relationship of coal mass could be expressed as

$$F(u) = \lambda u \exp \left[- \left(\frac{u}{u_c} \right)^m \right], \quad (1)$$

where λ represents the initial stiffness factor, u_c represents the maximum displacement, and m represents the kin exponent.

Reference:

[1]Karcinovic D. (1982): Statistical aspects of the continuous damage theory. International Journal of Solids and Structures, vol. 18, no. 7, pp. 551-562.

[2]Pan, Y.; Wang, Z. Q. (2004): Fold catastrophe model of rockburst in narrow coal pillar. Rock and Soil Mechanics, vol. 25, no. 1, pp. 23-30.

[3]Pan, Y.; Wang, Z. Q. (2004): Research approach on increment of work and energy-catastrophe theory of rock dynamic destabilization. Chinese Journal of Rock Mechanics and Engineering, vol. 23, no. 9, pp. 1433-1438.

3. In section 4.1, what are those parameters for?

A: Thank you for this suggestion. Those parameters in section 4.1 are prepared for the next numerical simulation in section 4.2 and theoretical calculation in section 4.6, for example, the intensity of this rock burst, in which the energy released is $6.4512 \times 10^7 \text{ J}$, is calculated by those parameters in section 4.1.

4. In section 4.2, rock burst index is a very effective index to predict rock burst tendency. How are the dynamic damage time, elastic energy index and impact energy index defined and measured?

A: Thank you for this suggestion. The burst tendency is an inherent property of coal, and there are four indices of burst tendency (dynamic damage time, uniaxial compressive strength, elastic energy index and impact energy index), which can be obtained in the complete stress–strain curve in mechanical test of bursting tendency of coal samples, with reference to the “classification and laboratory test method on bursting liability of coal (GB/T 25217.2-2010)”. To be more specific, the dynamic

damage time is the duration of the rapid drop in post-peak stress, and elastic energy index is the ratio of elastic deformation energy and dissipation deformation energy (the area of the hysteresis loop formed by the loading and unloading curve while the coal samples load from 75% to 85% of the peak strength to completely unload), and impact energy index is the ratio of the pre-peak stress accumulated deformation energy and the post-peak dissipated energy.

5. In section 4.4, the variation of stress abutment distance ahead of the working face (from 10 m at 50 m advance to 40 m at 100 m advance) should not be expected, in that the model which is homogeneous should yield similar volumes of failure zones around the working face.

A: Thank you for this suggestion. This manuscript was revised with reference to this suggestion.

6. The theory proposed is reasonable and innovative. Therefore, I recommend the use of theoretical models in numerical models and the authors can strengthen of the connection with the theoretical part.

A: Thank you for this suggestion. In mining practice, the width of coal pillar in gob-side entry is approximately 4 m to 10 m, and the length of working face is approximately 200 m to 250 m; thus, confined to size discrepancy, the stress evolution of coal seam mining is researched carefully by numerical simulation in this paper. Besides, the better understanding of stress and energy evolution of coal seam mining is beneficial for researching energy characteristics of coal pillar in gob-side entry. The more closely research related to the topic of the article would be done in follow-on work. This manuscript was revised with reference to this suggestion.

7. Coal and gas outburst and rock burst are the most serious dynamic disasters in coal mining. In shallow mining, most of the dynamic disasters are single coal and gas outburst or rock burst, and the interaction and interaction are not significant. With the increase of mining depth, the interaction between the two kinds of disasters began to

appear, resulting in coal and gas outburst, rock burst and two kinds of disasters coexisting and compounding each other. Both of them are unstable sudden dynamic disasters, and their triggering mechanism can be explained by energy theory. Whether the author tries to combine them?

A: Thank you for this suggestion. The mining practice indicates that interaction between two kinds of disasters (coal and gas outburst, rock burst) begins to appear in deep mining, and both of them are unstable sudden dynamic disasters. Two kinds of disasters (coal and gas outburst, rock burst) in coal mines represent the dynamic disaster of surrounding rocks in working face or during roadway excavation, and they are caused by the abrupt release of elastic strain energy in coal mass. Therefore, the energy theory could be adopted to reveal their triggering mechanism, which is meaningful and complex. In coal mining process, the excavation causes stress field redistribution in the original rock mass, and this phenomenon leads to stress concentration and elastic strain energy accumulation. It is obvious that the strength of a rockburst is proportional to the differential value of accumulated energy and dissipated energy in a coal mass. The total accumulated energy in the coal mass and the position of accumulated energy are vital to rockburst behavior. This research work would be carried out in follow-up work. This manuscript was revised with reference to this suggestion.

Reference:

[1]Pan YS (2016) Integrated study on compound dynamic disaster of coal-gas outburst and rockburst. *Journal of China Coal Society* 41(1): 105-112.

[2]Zhu LY, Pan YS, Li ZH, et al. (2018) Mechanisms of rockburst and outburst compound disaster in deep mine. *Journal of China Coal Society* 43(11): 3042-3050.

8. In the process of mining, the effects of soft coal and hard coal seem to be very different, and their risk of rock burst is different. For the model of Fig. 2a, does the author consider the influence of strength on rock burst of coal body?

A: Thank you for this suggestion. In Fig. 2a, the difference between soft coal and hard coal is the curve slope of F-u relationship of coal in Fig. 2b, and the influence of

strength on rock burst of coal body has been considered in the manuscript. This manuscript was revised with reference to this suggestion.

(a) interaction relationship of pillar-roof system

(b) load-displacement curve of mechanical system

Figure 2. Mechanical model of roadway roof and coal pillar.

9. The mining stress path is a process in which the vertical stress increases and the horizontal stress decreases during the mining process, which is inconsistent with the conventional stress loading path. Therefore, can Figure 3 reflect the effect of mining process on rock stress and strain? I suggest that the author explain this matter more clearly in order to provide better guidance and help for mining.

A: Thank you for this suggestion. In mining practice, the vertical stress increases and the horizontal stress decreases in the front of mining face, which is different from the conventional stress loading path. This stress state of coal has an important influence

on the mechanical response of coal, more importantly, the magnitude and location of peak stress of abutment pressure should be researched systematically. In Figure 3, the change rule of vertical stress (especially the characteristic of peak stress of abutment pressure) can be reflected in the manuscript. On this basis, better guidance and help for mining, such as reasonable development layout and protecting seam mining technology, are adopted in engineering practice to reduce the high stress concentration in the coal seam and prevent high load from leading the coal pillar to the peak stress point. Water injection in coal seam is also employed to reduce the strain-softening stiffness factor of the coal pillar in the post-peak stage and improve the stiffness factor of a double-block system in rock burst prevention. This manuscript was revised with reference to this suggestion.

Figure 3. Energy variation regularity in the mechanical model of roadway roof and coal pillar.

Reference:

[1]Xie HP, Zhou HW, and Liu JF, et al.(2011) Mining-induced mechanical behavior in coal seams under different mining layouts. Journal of China Coal Society, 36(7): 1067-1074.

Reviewer #2:

The authors present a theoretical analysis of the energy accumulation and dissipation of coal pillar in gob-side entry. The numerical simulation and case study are conducted to verify the rationality of the research method. I have a few major comments as follows:

1. In section 2.2, the author used two mathematical models, named "Weibull distribution law" and "damage theory of micro-statistics". What is the basis for this? A theoretical model established needs to be based on a large number of experiments. The authors need to justify that these two models can truly describe the strain softening characteristics of coal samples. Otherwise, the results derived from such a non-consistent model are thus questionable.

A: Thank you for this suggestion. In the uniaxial compression test of coal sample, when the working load reaches the compressive strength of coal sample, the ability of coal sample to resist deformation weakens with the increment of compression amount. The property of coal sample is known as the strain softening characteristics in post-peak stage. As shown in the stress-strain curve of coal sample in figure (a), when $d\varepsilon > 0, d\sigma < 0,$ and $\sigma'(\varepsilon_0) < 0$ in the post-peak stage.

Figure (a): the stress-strain curve of coal sample

Many experiments (such as Karcinovic D. (1982); Pan, Y. (2004) et al.) have been done to clarify that "Weibull distribution law" and "damage theory of micro-statistics"

are suitable for the strain softening characteristics of coal samples. On this basis, it is valid to use "Weibull distribution law" and "damage theory of micro-statistics" to describe the F-u relationship of coal, and the load–displacement relationship of coal mass could be expressed as

$$F(u) = \lambda u \exp \left[- \left(\frac{u}{u_c} \right)^m \right], \quad (1)$$

where λ represents the initial stiffness factor, u_c represents the maximum displacement, and m represents the kin exponent.

Reference:

- [1]Karcinovic D (1982): Statistical aspects of the continuous damage theory. International Journal of Solids and Structures, 18(7):551-562.
- [2]Pan Y; Wang ZQ (2004): Fold catastrophe model of rockburst in narrow coal pillar. Rock and Soil Mechanics, 25(1):23-30.
- [3]Pan Y; Wang ZQ(2004): Research approach on increment of work and energy-catastrophe theory of rock dynamic destabilization. Chinese Journal of Rock Mechanics and Engineering, 23(9):1433-1438.

2. What is "quasi-static deformation". Please explain it in detail.

A: Thank you for this suggestion. In this article, the quasi-static deformation is the mechanical response of coal pillar under the action of quasi-static loading, namely, the term of inertia force caused by deformation is omitted. This manuscript was revised with reference to this suggestion.

3. How to determine the position of the points j and s in the coal pillar instability stage (Figure 3)?

A: Thank you for this suggestion. In Figure 3, the slope of the post-peak stage of coal sample in j and s points are equal to that of oblique *on*, the first point is j , and the next point is s ; thus, the initial point and terminal point of dynamic destabilization of coal sample is point j and point s , respectively. This manuscript was revised with

reference to this suggestion.

Figure 3. Energy variation regularity in the mechanical model of roadway roof and coal pillar.

4. The authors need to justify the reasonability of the fold catastrophe model compared to other previous studies.

A: Thank you for this suggestion. In previous studies, the catastrophe model is suitable to clarify the dynamic destabilization, and the cusp catastrophic model is usually employed to analyze the instability process, such as the destabilization of geometry form of the elastic body. Compared to the cusp catastrophic model, the fold catastrophe model is consistent with the key characteristics of dynamic destabilization of coal pillar, such as the irreversibility of destabilization, two types of steady state (unstable equilibrium in point j and stable equilibrium in point s in Figure 3).

Figure 3. Energy variation regularity in the mechanical model of roadway roof and coal pillar.

The fold catastrophe model is one of seven types of catastrophe mathematical model. According to the fold catastrophe theory, the control variable a and state variable

x are both one, the potential function is $\frac{x^3}{3} + ax$, and the equilibrium surface equation is $x^2 + a = 0$. The point in equilibrium surface refers to an equilibrium state of mechanical system, and the equilibrium surface of fold catastrophe model is shown in Fig. 5.

Figure 5: Balance curve of the fold catastrophe model

In equilibrium surface equation of fold catastrophe model, x is the state variable, and a is the control variable. The system is in dummy status if $a > 0$, while the system is a parabola if $a \leq 0$; the straight line $a = 0$ (or the axis $K - 1$) divides the parabola into upper and lower branches. x_1 and x_2 represent the expression of state variable x on branch 1 and branch 2, respectively. In branch 1, $x_1 < 0$, which refers to the unstable equilibrium, and in branch 2, $x_2 > 0$, which refers to the stable equilibrium. Therefore, there are two equilibrium states for the fold catastrophe model; one is unstable, and the other is stable; besides, one equilibrium state cannot change to the other equilibrium state. This manuscript was revised with reference to this suggestion.

5. In line 35 of P11, what is the criterion for judging the coal seam's burst tendency?

A: Thank you for this suggestion. The burst tendency is an inherent property of coal,

and the criterion for judging the coal seam's burst tendency is “classification and laboratory test method on bursting liability of coal (GB/T 25217.2-2010)”. To be more specific, there are four indices of burst tendency (dynamic damage time, uniaxial compressive strength, elastic energy index and impact energy index), which can be obtained in mechanical test of bursting tendency of coal samples, and these all play an important role in evaluating rock burst hazards in coal seams. In this article, the dynamic damage time is 46 ms (strong), the elastic energy index is 6.79 (strong), the impact energy index is 5.14 (strong), and the uniaxial compressive strength is 14.0 MPa (strong), according to the bursting tendency classification of coal sample, shown in table 1. Therefore, the comprehensive evaluation results of burst tendency indicate that the coal samples show a strong impact tendency. This manuscript was revised with reference to this suggestion.

Table 1. The bursting tendency classification of coal sample

type	I	II	III
bursting liability	none	weak	strong
dynamic damage time /ms	$DT > 500$	$50 < DT \leq 500$	$DT \leq 50$
elastic energy index	$W_{ET} < 2$	$2 \leq W_{ET} < 5$	$W_{ET} \geq 5$
impact energy index	$K_E < 1.5$	$1.5 \leq K_E < 5$	$K_E \geq 5$
uniaxial compressive strength /MPa	$R_c < 7$	$7 \leq R_c < 14$	$R_c \geq 14$

6. Page 13. Correct: Figures 11 and 12.

A: Thank you for this suggestion. This manuscript was revised with reference to this suggestion.

7. In the numerical modeling, I didn't see any results about crack propagation. How does the author come to this conclusion that “Many cracks in coal seam are development,”? (Please see Wu et al., 2019. Rock Mechanics and Rock Engineering)

A: Thank you for this suggestion. The conclusion that “Many cracks in coal seam are developed,” is obtained in mining practice, rather than in numerical simulation.

This manuscript was revised with reference to this suggestion.

8. In figure 14, the energy density at the two ends of goaf reach to 1.96×10^{16} J. This value is so large. Is it reasonable? Please further analyze and discuss.

A: Thank you for this suggestion. In this article, the unit of energy density in figure 14 is J/mm^3 , so the energy density at the two ends of goaf reaches to $1.96 \times 10^{16} \text{ J}/\text{mm}^3$, namely $1.96 \times 10^7 \text{ J}/\text{m}^3$. This manuscript was revised with reference to this suggestion.

9. I didn't see the roadway driving along the goaf in the numerical model. Therefore, how does the author compare the theoretical results with the observation result (Line 14 to 16 of P15)? It is questionable.

A: Thank you for this suggestion. In this article, the numerical simulation focuses on the analysis the stress evolution of coal seam mining rather than roadway driving along goaf, and theoretical results is compared with the observation result in mining practice. To be more precise, in mining practice, a rock burst induced by coal pillar in the gob-side entry, located in frontal abutment pressure region of No. 3306 working face, occurred in Xin'an coal mine. The earthquake observation station records that the rock burst intensity is equivalent to an earthquake of 2.3 Richter magnitude (in section 4.1). Meanwhile, in theoretical analysis, The intensity of this rock burst, in which the energy released is $6.4512 \times 10^7 \text{ J}$, is equivalent to an earthquake of 2.1 Richter magnitude (in section 4.6). Therefore, the theoretical result is close to the observation result in mining practice. This manuscript was revised with reference to this suggestion.

10. The topic of paper is energy characteristics of coal pillar in gob-side entry. But, in the section of numerical simulation, the authors focuses on the analysis the stress evolution of coal seam mining rather than roadway driving along goaf. The numerical results need to be more closely related to the topic of the article.

A: Thank you for this suggestion. In mining practice, the width of coal pillar in gob-side entry is approximately 4 m to 10 m, and the length of working face is

approximately 200 m to 250 m; thus, confined to size discrepancy, the stress evolution of coal seam mining is researched carefully by numerical simulation in this paper. Besides, the better understanding of stress and energy evolution of coal seam mining is beneficial for researching energy characteristics of coal pillar in gob-side entry. The more closely research related to the topic of the article would be done in follow-on work.